# Diagnostic Value of Multi-Mode Ultrasonic Flow Imaging Examination in Solid Renal Tumors of Different Sizes

**DOI:** 10.3390/jcm12020566

**Published:** 2023-01-10

**Authors:** Dai Zhang, Ying Wang, Fan Yang, Yiran Mao, Jie Mu, Lihui Zhao, Wengui Xu

**Affiliations:** 1Department of Diagnostic and Therapeutic Ultrasonography, Tianjin Medical University Cancer Institute and Hospital, Tianjin 300060, China; 2National Clinical Research Center for Cancer, Tianjin 300060, China; 3Key Laboratory of Cancer Prevention and Therapy, Tianjin 300060, China; 4Tianjin’s Clinical Research Center for Cancer, Tianjin 300060, China; 5Department of Molecular Imaging and Nuclear Medicine, Tianjin Medical University Cancer Institute and Hospital, Tianjin 300060, China

**Keywords:** renal tumor, MFI, CDFI, PDI

## Abstract

Purposes: To explore the value of Microflow Imaging (MFI) in renal solid tumors. Methods: A total of 195 patients with 199 lesions pathologically confirmed masses were retrospectively analyzed. The 199 masses were divided into the tumor ≤ 4 cm group (*n* = 104) and tumor > 4 cm group (*n* = 95). The diagnostic efficacy of Color Doppler Flow Imaging (CDFI), Power Doppler Imaging (PDI) and MFI in renal tumors sizes were compared by determining the Adler grade, vascular morphology and peripheral blood flow. Results: Among 199 tumors, 161 lesions were malignant and 38 lesions were benign. MFI in malignant tumor ≤ 4 cm demonstrated statistically significant differences in Adler grade and vascular morphology as compared to CDFI and PDI (*p* < 0.05). In malignant tumor > 4 cm group, MFI showed significant difference in vascular morphology compared with CDFI (*p* < 0.05). MFI showed a significant difference in the peripheral annular blood flow of malignant tumors when compared to CDFI and PDI (*p* < 0.05). In addition, the malignant tumors of the two sizes by MFI in peripheral annular blood flow detection showed significant difference (*p* < 0.05). The area under the curve of ROC by MFI in the tumor ≤ 4 cm was 0.771, which was higher than CDFI and PDI (*p* < 0.05), but no obvious difference among the tumor > 4 cm (*p* > 0.05). Conclusion: MFI provides a new method for the differential diagnosis of small renal carcinoma. Based on the convenience and non-radiation of MFI, we can choose MFI as an imaging diagnostic tool for patients who need long-term active surveillance (AS) follow-up.

## 1. Introduction

Clinically, renal tumors with a diameter ≤ 4 cm are called small renal masses [1]. With the increasing attention of people to physical examination, its detection rate has also been greatly increased. At the same time, with the change of management mode of small renal masses, the method of radical nephrectomy has changed to nephron-sparing approaches, including AS [2,3]. In 2017, the American Urological Association (AUA) guidelines recommended AS conduct an acceptable choice for all patients with kidney tumors ≤ 2 cm, and a prospective study suggests that the management of AS may also be appropriate for some renal tumors ≤ 4 cm [4,5]. Therefore, accurate preoperative assessment of benign and malignant small renal tumors is very important for clinical decision-making”.

Kidney cancer is rich in microvessels. The blood flow grading and vascular morphology are essential to determine the nature of the lesion [6]. CDFI and PDI are useful in evaluating tumor vascularity, which, however, is restricted in detecting the blood flow of low-velocity and microvasculature [7,8,9]. MFI adopts an adaptive algorithm, and is informative for assessing low-speed blood flow state [10]. To investigate the value of MFI in evaluating renal solid tumors of different sizes vessel characteristics and in differentiating benign from malignant tumors, we retrospectively compared the vascular characteristics between MFI, PDI and CDFI.

## 2. Materials and Methods

### 2.1. Study Population

This study was approved by Tianjin Medical University Cancer Institute and Hospital ethics committee, and all the patients signed the informed consent. All the data of patients in this study were consecutive. The clinical data of patients with pathologically confirmed renal mass from June 2018 and August 2021 in the hospital were retrospectively analyzed. Inclusion criteria: All patients were examined by CDFI, PDI and MFI. Patients ≥ 18 years old who did not received chemotherapy and had complete pathological data. Exclusion criteria: Patients < 18 years old, with no clear pathologic findings or prior preoperative treatment. All patients underwent grayscale sonography, CDFI, PDI and MFI examinations, and surgical treatment to obtain pathological results.

### 2.2. Instruments and Methods

A Philips EPIQ5 ultrasound diagnostic scanner was used with a 1–5 MHz convex probe. The instrument is equipped with CDFI, PDI and MFI functions. Baseline grayscale ultrasound scans were first obtained. The main findings were recorded, including the location, size, boundary, echo, homogeneity, small cystic area and other basic information. Subsequently, scans were repeated with the following modes: CDI, PDI, MFI, to observe the vasculature within the mass. Gain settings were optimized for each imaging techniques. The scanning was performed while each patient held their breath. The region of interest (ROI) involved the tumor and its adjacent normal tissues (1 cm). Synchronous static and dynamic images were stored.

### 2.3. Imaging Analysis

The blood flow grading, vascular morphology and peripheral blood flow of the renal mass were analyzed. According to Adler’s method, the blood flow grading was determined [11,12]: Grade 0: no blood flow in the lesion; Grade 1: 1~2 pixels contained blood flow (usually <1 mm in diameter) was observed in the lesion; Grade 2: 3~4 pixels or a main vessel were visualized in the lesion; Grade 3: ≥5 pixels or ≥2 main vessels were visualized inside the lesion. Vascular morphology includes [13]: (1) None, without any clear blood flow signal; (2) Dotted or linear, in which the blood flow signal could be detected as the shape of dot or line; (3) Dendritic: arborization with blood flow distributed in arborization within tumor; (4) Irregular, in which the tortuously and disorderedly distributed blood flow was detected in tumor. Peripheral blood flow was recorded as annular and annulus free blood flow.

Image interpretation was completed by two radiologists (approximately 10 years of clinical experience in sonography, CDFI, PDI and 3 years in MFI), and each was unaware of each other’s diagnosis and pathological results. When the diagnosis was inconsistent, they discussed the diagnosis results and reached an agreement. Benign tumors were characterized by Adler grade 0~1, vascular morphology was none or dotted or linear, annulus free peripheral blood flow signal, combined with grayscale ultrasound information. Malignant tumors were characterized by Adler grade 2~3, vascular morphology was dendritic or irregular, and peripheral annular blood flow signal, combined with grayscale ultrasound information. All lesions were surgically resected to obtain histological and pathological results, which were used as the gold standard for the diagnosis of benign and malignant renal masses.

### 2.4. Statistical Methods

Statistical analysis was performed by SPSS 23.0 (SPSS Inc, Chicago, IL, USA). Data were presented as mean ± standard deviation for continuous variables and frequencies with percentages for categorical variables. The comparison of categorical variables between the two groups was performed using the χ2 test. Kappa test was used for consistency between different methods. Among them, kappa ≥ 0.75 can be considered to have good consistency; 0.4 < kappa < 0.75, indicating medium consistency; kappa ≤ 0.4 indicates poor consistency. The sensitivity, specificity, positive likelihood ratio (+LR), negative likelihood ratio (−LR), positive predictive value, negative predictive value and accuracy were calculated between the two groups. The software MedCalc was used to draw the ROC curve, and the area under the curve (AUC) was compared using the Z test. *p* < 0.05 was considered statistically significant.

## 3. Results

### 3.1. Pathological Results

A total of 199 lesions from 195 patients were included: 118 patients were males and 77 were females, ranging from 26 to 82 years old, with an average of (56.21 ± 10.6) years old. There were 104 masses with a largest diameter of 0–4 cm, and 95 masses with a largest diameter of >4 cm. According to the pathological results, 38 cases were benign, and 161 cases were malignant. The detailed pathological pattern is shown in Table 1.

### 3.2. Grayscale Ultrasonic Features

The grayscale ultrasound characteristics of renal tumors in this study are shown in Table 2. There was no significant difference in location, boundary, echo, homogeneity and small cystic area between malignant and benign tumors ≤ 4 cm (*p* > 0.05); In renal tumor > 4 cm group, there were statistically significant differences in echo and small cystic area between malignant and benign tumors (*p* < 0.05) (Figure 1A, Figure 2A, Figure 3A and Figure 4A).

### 3.3. CDFI, PDI and MFI Evaluation of Blood Flow Characteristics in Benign and Malignant Renal Tumors ≤ 4 cm

Table 3 demonstrates that in benign tumors ≤ 4 cm, there is no significant difference between MFI and CDFI and between MFI and PDI in Adler grade, vascular morphology and peripheral blood flow (*p* > 0.05). In malignant tumors, MFI showed statistically significant differences with CDFI and PDI in Adler grade, vascular morphology and peripheral blood flow detection (*p* < 0.05); of a CDFI and PDI in these areas have no obvious difference (*p* > 0.05) (Figure 1B–D and Figure 2B–D).

### 3.4. CDFI, PDI and MFI Evaluation of Blood Flow Characteristics in Benign and Malignant Renal Tumors > 4 cm

Table 4 shows that in benign tumors > 4 cm, between MFI and CDFI and between MFI and PDI there is no statistical significance in Adler grade, vascular morphology and peripheral blood flow (*p* > 0.05). In malignant tumors, MFI had statistically significant differences with CDFI and PDI in peripheral blood flow detection (*p* < 0.05). There was a significant difference in vascular morphology between CDFI and MFI (*p* < 0.05) but no significant difference between PDI and MFI (*p* > 0.05). In terms of Adler grade, MFI had no obvious difference compared with of a CDFI and PDI (*p* > 0.05) (Figure 3B–D and Figure 4B–D).

### 3.5. Peripheral Blood Flow Characteristics of Renal Tumor in MFI Mode

For benign renal tumor smaller than 4 cm and larger than 4 cm, the detection rate of peripheral annular blood flow by MFI was 16.67% and 5%, respectively, and there was no significant difference between them (χ2 = 1.369, *p* = 0.242). For malignant renal tumor smaller than 4 cm and larger than 4 cm, the detection rate of peripheral annular blood flow by MFI was 82.56% and 57.33%, respectively, and the difference between them was statistically significant (χ2 = 12.332, *p* = 0.000).

### 3.6. Evaluation of Diagnostic Result by CDFI, PDI and MFI in Benign and Malignant Renal Tumors of Different Sizes

The diagnostic result of CDFI, PDI and MFI in renal tumors of different sizes were shown in Table 5. In tumors ≤ 4 cm, the diagnosis consistency between CDFI and PDI was good (Kappa = 0.936, *p* = 0.000), and the diagnosis consistency between MFI and CDFI was moderate (Kappa = 0.491, *p* = 0.000). The consistency between MFI and PDI was moderate (Kappa = 0.556, *p* = 0.000). In tumors > 4 cm, the diagnosis consistency between CDFI and PDI was good (Kappa = 0.833, *p* = 0.000), and the diagnosis consistency between MFI and CDFI was moderate (Kappa = 0.699, *p* = 0.000). The diagnosis of MFI was consistent with that of PDI (Kappa = 0.804, *p* = 0.000). For renal tumors ≤ 4 cm, the areas under the ROC curve of the diagnostic efficacy by CDFI, PDI and MFI were 0.694, 0.700 and 0.771, respectively (Figure 5). The Z test results showed that the diagnostic efficacy of MFI compared with CDFI and PDI was statistically significant (Z_1_ = 2.167, *p*_1_ = 0.0302; Z_2_ = 2.020, *p*_2_ = 0.0434), but there was no significant difference between CDFI and PDI (Z = 1.000, *p*= 0.3173). For renal tumors > 4 cm, the areas under the ROC curve were 0.802, 0.797 and 0.805, respectively (Figure 6). The Z test results showed no statistical difference in the diagnostic efficacy between MFI, CDFI and PDI (Z1 = 0.182, *p*_1_ = 0.8556; Z2 = 0.0859, *p*_2_ = 0.9316; Z3 = 0.288, *p*_3_ = 0.7731).

## 4. Discussion

At present, gray-scale ultrasound has become the preferred imaging examination technology for the diagnosis of renal tumors, which plays an important role in the early detection of renal tumors and the detection and diagnosis of small renal cell carcinoma in physical examination. According to statistics, more than ⅔ of the early lesions are found incidentally by ultrasound [4]. However, because the ultrasonographic images of renal tumors often overlap, there are different lesions with the same shadows or the same lesions with different shadows. Gray-scale ultrasound and color Doppler ultrasound are susceptible to factors such as location and detection angle of the lesions, and the detection rate of low-speed blood flow is low. which cannot show the perfusion of the microcirculation of the lesions. Therefore, ultrasound has some limitations in the qualitative diagnosis of renal tumors.

Contrast-enhanced ultrasound (CEUS) can obtain enhanced features of renal masses, better display the internal and peripheral enhancement of the masses, and help to predict the benign and malignant renal masses and analyze the histological subtypes of malignant renal tumors [14], but it requires a high level of operation by the operator, and patients are at high risk of allergy and high cost.

CT-scan has a high accuracy in locating and qualitatively evaluating renal tumors. It has obvious advantages in evaluating signs of retroperitoneal lymph node metastasis, perirenal lymph node infiltration, renal vein and inferior vena cava cancer thrombus. It has always been considered as the gold standard in imaging diagnosis. However, for patients with a diameter ≤ 4 cm, the diagnosis of renal carcinoma with internal hemorrhage, necrosis, cystic change or some complicated cysts with little blood supply, it is easy to be confused with some benign tumors [15]. Moreover, CT-scan is radiative, which is not conducive to be used as an imaging tool for long-term follow-up.

In this study, 51.16% of the malignant tumors ≤ 4 cm were hyperechoic; 50.1% of the benign tumors ≤ 4 cm were hypoechoic, and 77.78% of them were heterogeneous, and one patient had a small cystic area. This is not consistent with the previous conclusion of ‘Malignant tumors were mainly hypoechoic, benign tumors were hyperechoic and homogeneous. ’The results showed that there was no significant difference between malignant and benign tumors ≤ 4 cm in location, boundary, echo, homogeneity and small cystic area (*p* > 0.05). However, there were statistically significant differences in echo and small cystic area between benign and malignant tumors > 4 cm (*p* < 0.05); no significant difference in other aspects (*p* > 0.05). Therefore, grayscale ultrasound alone cannot be used to determine the tumor properties, especially for the renal tumors ≤ 4 cm.

Angiogenesis is an essential part of the tumor microenvironment [16]. Abnormal angiogenesis is an important characteristic of malignant tumors [17,18], while renal tumors of different sizes and types show different internal blood flow morphology and distribution.

MFI is a new ultrasound imaging technology, which can detect blood vessels with a diameter of 0.1 mm and a flow rate of 1 cm/s. It is more effective than CDFI and PDI in microvascular visualization [9]. At present, some scholars had applied MFI technology to the diagnosis of liver cancer and breast cancer [19,20], but there are few reports on the application of MFI technology in renal cell carcinoma.

Benign tumors have relatively slow angiogenesis, relatively regular blood vessels and fewer branches [21]. In this study, benign tumors regardless of size, had relatively low blood flow grading, and vascular morphology was common in dotted or liner, and peripheral blood flow was rarely explored. There were no significant differences in Adler grade, vascular morphology and peripheral blood flow of benign tumors between CDFI, PDI and MFI, which was consistent with the results of MAO et al. [13]. MFI can detect tiny and low-speed blood flow signals. Some benign masses with abundant blood flow were prone to misdiagnosis. The specificity of MFI for the diagnosis of benign masses in the two groups was 61.11% and 65%, respectively, both lower than CDFI and PDI. The negative predictive value of MFI for the diagnosis of benign masses in the two groups was 64.71% and 81.25%, respectively, both higher than CDFI and PDI. However, the Negative likelihood Ratio was 0.11 and 0.06, respectively, which indicated that although the application of MFI would cause a certain misdiagnosis, the diagnosis of benign tumors by MFI had a certain value. In this study, for the tumors smaller than 4 cm, there were 3 cases with irregular vascular morphology and 50% of them had Adler grade 2~3 by MFI. For the tumors larger than 4 cm, there were three cases with dendritic vascular morphology and four cases with irregular vascular morphology, and 55% of them had Adler grade 2~3 by MFI. Therefore, for benign masses with high blood flow grading, the diagnosis accuracy can be improved if the analysis is combined with the vascular morphology and grayscale ultrasonic characteristics.

In malignant tumors, there is a great genesis of blood vessels that are featured with irregular and wide diameters, deformation, distortion and even arteriovenous fistula [17,18]. In this study, malignant tumors had relatively high blood flow grading, and vascular morphology was common in irregular or branch, and peripheral blood flow was regularly explored. While, there were significantly different between MFI and CDFI, as well as between MFI and PDI in the evaluation of blood flow characteristics of malignant tumors ≤ 4 cm. Using CDFI and PDI, most malignant tumors ≤ 4 cm were Adler grade 1 and grade 3, and the vascular morphology was mainly dotted or linear, 27.91% and 36.05% of cases demonstrated peripheral annular blood flow. This pattern of blood flow is similar to previous studies [22]. Examination of malignant tumors ≤ 4 cm using MFI revealed the Adler grade was mainly grade 3, with an irregular vascular morphology, peripheral annular blood flow was detected in 82.56% of cases. This indicates that small tumors feature low vascularization and slow blood flow, which are appropriate conditions for MFI to identify low-speed blood flow signals and delicate vascular branches that CDFI and PDI cannot recognize. For malignant renal tumors > 4 cm, as the accretion of the tumor, the vascular growth factor of the tumor stimulates the generation of new capillaries and increases the number of blood vessels. In addition, larger tumors are richer in blood vessels and lymphatic network, and a higher number of new small vessels are observed. CDFI and PDI can be used to explore rich, twisted blood vessels, so the advantage of MFI is weakened.

Renal cell carcinoma is often surrounded by pseudocapsule, which lies between normal renal tissue and the tumor, containing capillary vessel and fibrous tissue [23]. Ultrasonography usually shows the annular blood vessels around the tumor, which is the characteristic of malignant tumor. In this study, CDFI and PDI had low detection rate of peripheral annular blood flow in renal malignancies of different sizes, while MFI had higher detection ability. Further analysis of revealed that the peripheral annular blood flow detection rate in the malignant tumor ≤ 4 cm was higher than that the tumor > 4 cm. This may be due to the moderate tumor volume, suitable growth time, complete vascular circle and easy detection by MFI in smaller tumors. However, the tumor > 4 cm had a higher degree of malignancy. Its rapid growth rate led to the destruction of capillary rings around the tumor, so it could not be detected. In addition, due to the large size of the tumor, part of the tumor was located deeper in abdominal cavity, and MFI could not detect the blood flow signal [24].

The limitations of this study are as follows: 1. This is a retrospective study with statistical bias. Prospective multicenter analyses can provide more information and reduce selection bias. 2. MFI technology is still being researched and image interpretation standards have not yet been developed. 3. This study analyzed only the difference in blood flow characteristics of benign and malignant renal tumors using the MFI technique and did not analyze the different subtypes of renal cell carcinoma, which will be the next research focus.

## 5. Conclusions

MFI can effectively detect the low-speed blood flow and small blood vessels of renal tumors and improve the peripheral annular blood flow display rate of malignant tumors. Especially in tumors smaller than 4 cm, the blood flow detection of MFI is superior to CDFI and PDI. MFI providing a new method for the differential diagnosis of small renal cancer. Based on the convenience and non-radiation of MFI, we can choose MFI as the imaging diagnostic tool for patients who need AS for long-term follow-up.

## Figures and Tables

**Figure 1 jcm-12-00566-f001:**
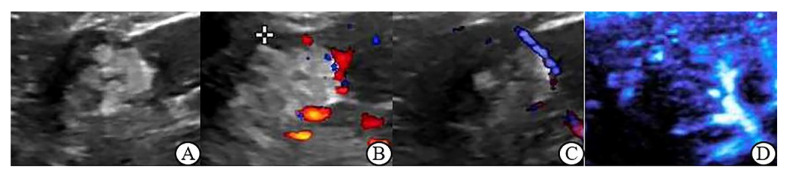
Angiomyolipoma (largest diameter: 2.9 cm) (**A**): Grayscale ultrasound: the mass with clear boundary, hyperechoic, and the internal echogenicity was heterogeneous; (**B**): CDFI: 2 dotted blood flow signal was observed in the lesion, Adler Grade 1; (**C**): PDI: Linear blood flow signal was seen around the lesion, Adler Grade 1; (**D**): MFI: 2 dotted blood flow signal was observed in the lesion, 2 linear blood flow signal was observed around the lesion, Adler Grade 2, there is no annulus blood flow.

**Figure 2 jcm-12-00566-f002:**
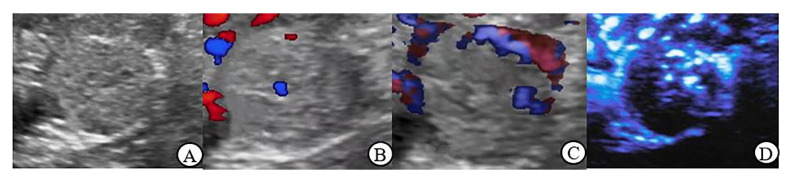
Clear cell carcinoma (largest diameter: 2.1 cm) (**A**): Gray scale ultrasound: the mass with clear boundary, hypoechoic, and the internal echogenicity was heterogeneous; (**B**): CDFI: 1 dotted blood flow signal was observed in the lesion, Adler Grade 1; (**C**): PDI: 4 dotted blood flow signal was observed in the lesion, Adler Grade 2; (**D**): MFI: rich dotted was observed in the lesion, Adler Grade 3, peripheral annular blood flow was recorded.

**Figure 3 jcm-12-00566-f003:**
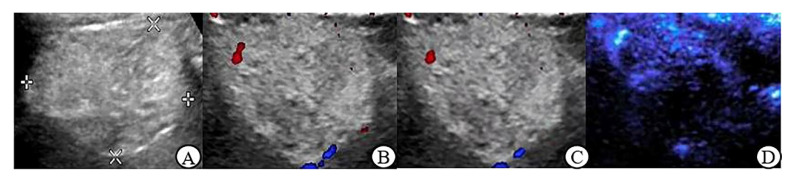
Angiomyolipoma (largest diameter 5.7 cm) (**A**): Gray scale ultrasound: the mass with clear boundary, hyperechoic, and the internal echogenicity was heterogeneous; (**B**): CDFI: 3 dotted blood flow signal was observed in the lesion, Adler Grade 2; (**C**): PDI: 2 dotted blood flow signal was observed in the lesion, Adler Grade 1; (**D**): MFI: dotted and linear blood flow signal was observed in the lesion, Adler Grade 2, there is no annulus blood flow.

**Figure 4 jcm-12-00566-f004:**
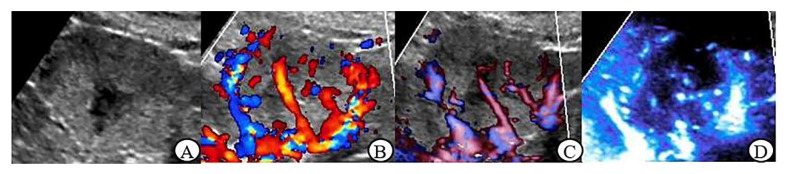
Clear cell carcinoma (largest diameter: 4.1 cm) (**A**): Gray scale ultrasound: the mass with clear boundary, hypoechoic, heterogeneous, and the cystic lesions was explored; (**B**): CDFI: irregular blood flow signal was observed in the lesion, Adler Grade 3, peripheral annular blood flow was recorded; (**C**): PDI: irregular blood flow signal was observed in the lesion, Adler Grade 3, there is no annulus blood flow; (**D**): MFI: irregular blood flow pattern within the lesion, Adler Grade 3, peripheral annular blood flow was not obvious.

**Figure 5 jcm-12-00566-f005:**
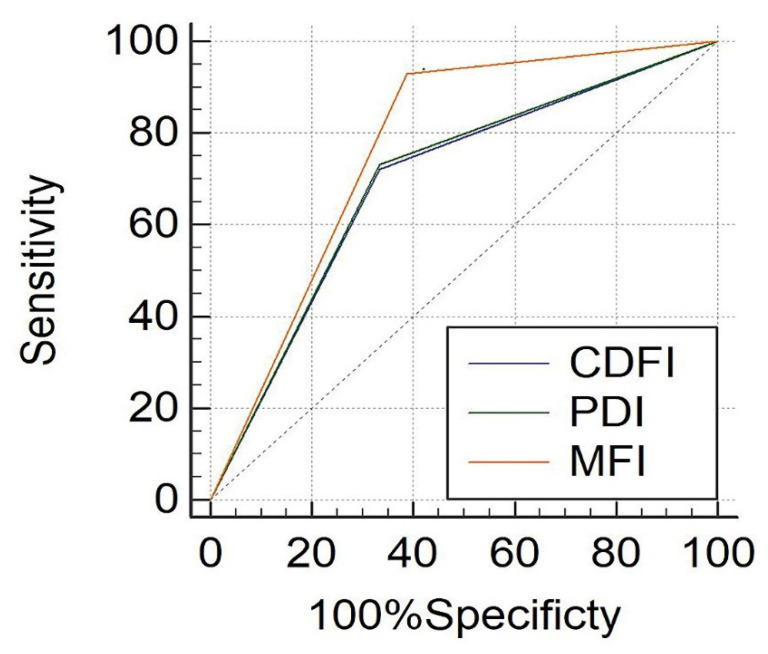
The ROC curve of CDFI, PDI and MFI for renal tumors with the largest diameter ≤ 4 cm.

**Figure 6 jcm-12-00566-f006:**
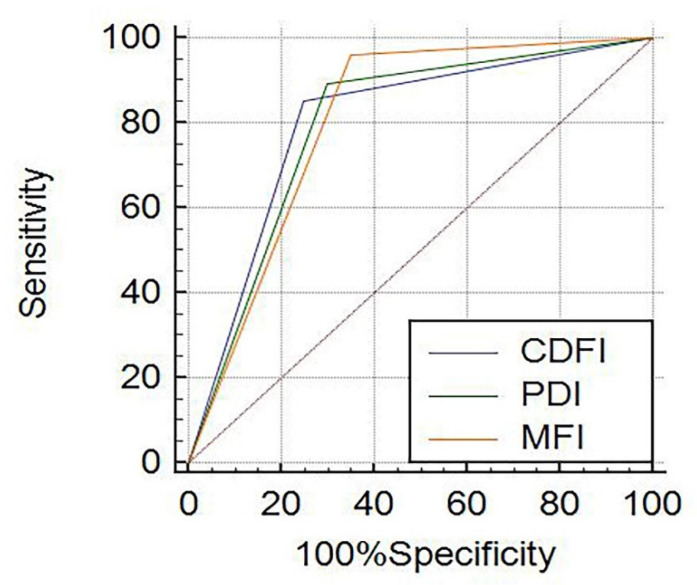
The ROC curve of CDFI, PDI and MFI for renal tumors with the largest diameter > 4 cm.

**Table 1 jcm-12-00566-t001:** Pathological type of registered patients.

Renal Tumor ≤ 4 cm (104)	Renal Tumor > 4 cm (95)
clear cell carcinoma (76)	clear cell carcinoma (61)
chromophobe cell carcinoma (7)	chromophobe cell carcinoma (10)
papillary cell carcinoma (3)	papillary cell carcinoma (4)
Angiomyolipoma (15)	Angiomyolipoma (14)
Oncocytoma (2)	Oncocytoma (3)
epithelioid-angiomyolipoma (1)	epithelioid-angiomyolipoma (3)

**Table 2 jcm-12-00566-t002:** Grayscale ultrasonic features of benign and malignant renal tumors in two Groups (n).

	Renal Tumor ≤ 4 cm (104)				Renal Tumor > 4 cm (95)			
	Benign (18)	Malignant (86)	χ2	*p*	Benign (20)	Malignant (75)	χ2	*p*
**Location**								
Left	7 (38.89%)	39 (45.35%)	0.252	0.616	8 (40%)	37 (49.33%)	0.552	0.458
Right	11 (61.11%)	47 (54.65%)			12 (60%)	38 (50.67%)		
**Boundary**								
Clear	17 (94.44%)	67 (77.91%)	2.621	0.105	15 (75%)	61 (81.33%)	0.396	0.529
Unclear	1 (5.56%)	19 (22.09%)			5 (25%)	14 (18.67%)		
**Echo**								
Hypoechoic	9 (50%)	42 (48.84%)	0.008	0.928	5 (25%)	63 (84%)	27.018	0.000
Hyperechoic	9 (50%)	44 (51.16%)			15 (75%)	12 (16%)		
**Internal echogenicity**								
Homogeneity	4 (22.22%)	25 (29.07%)	0.347	0.556	4 (20%)	5 (6.67%)	3.273	0.070
Heterogeneous	14 (77.78%)	61 (70.93%)			16 (80%)	70 (93.33%)		
**Cystic lesions**								
Yes	1 (5.56%)	23 (26.74%)	3.765	0.052	5 (25%)	42 (56%)	6.070	0.014
No	17 (94.44)	63 (73.26%)			15 (75%)	33 (44%)		

**Table 3 jcm-12-00566-t003:** Blood flow characteristics of renal benign and malignant tumors ≤ 4 cm evaluated by CDFI, PDI and MFI (n).

	CDFI		PDI		MFI	
	Benign (18)	Malignant (86)	Benign (18)	Malignant (86)	Benign (18)	Malignant (86)
**Adler grade**						
0	8 (44.44%)	12 (13.95%)	6 (33.33%)	11 (12.79%)	5 (27.78%)	3 (3.49%)
1	3 (16.67%)	25 (29.07%)	5 (27.78%)	21 (24.42%)	4 (22.22%)	5 (5.81%)
2	5 (27.78%)	20 (23.26%)	4 (22.22%)	18 (20.93%)	3 (16.67%)	15 (17.44%)
3	2 (11.11%)	29 (33.72%)	3 (16.67%)	36 (41.86%)	6 (33.33%)	63 (73.26%)
**Vascular morphology**						
None	8 (44.44%)	12 (13.95%)	6 (33.33%)	11 (12.79%)	5 (27.78%)	3 (3.49%)
Dotted or liner	10 (55.56%)	56 (65.12%)	11 (61.11%)	46 (53.49%)	10 (55.56%)	24 (27.91%)
Branch	0	5 (5.81%)	0	8 (9.3%)	0	19 (22.09%)
Irregular	0	13 (15.12%)	1 (5.56%)	21 (24.42%)	3 (16.67%)	40 (46.51%)
**Peripheral annular blood flow**						
None	18 (100%)	62 (72.09%)	18 (100%)	55 (63.95%)	15 (83.33%)	15 (17.44%)
Annular	0	24 (27.91%)	0	31 (36.05%)	3 (16.67%)	71 (82.56%)

Comments: Benign tumors: CDFI&PDI: χa2 = 1.097, pa = 0.778; χb2 = 0.305, pb = 0.581; χc2 = 0.000, pc = 1.000; MFI&CDFI: χa2 = 3.354, pa = 0.343; χb2 = 2.769, pb = 0.25; χc2 = 3.273, pc = 0.07; MFI&PDI: χa2 = 1.345, pa = 0.719; χb2 = 2.945, pb = 0.229; χc2 = 3.273, pc = 0.07. Malignant tumors: CDFI&PDI: χa2 = 1.250, pa = 0.741; χb2 = 3.599, pb = 0.308; χc2 = 1.310, pc = 0.252; MFI&CDFI: χa2 = 32.013, pa = 0.000; χb2 = 40.121, pb = 0.000; χc2 = 51.941, pc = 0.000; MFI&PDI: χa2 = 22.054, pa = 0.000; χb2 = 21.885, pb = 0.000; χc2 = 38.543, pc = 0.000. Where a represents Adler classification, b represents vascular morphology, c represents Peripheral blood flow.

**Table 4 jcm-12-00566-t004:** Blood flow characteristics of benign and malignant renal tumors > 4 cm evaluated by CDFI, PDI and MFI (n).

	CDFI		PDI		MFI	
	Benign (20)	Malignant (75)	Benign (20)	Malignant (75)	Benign (20)	Malignant (75)
**Adler grade**						
0	11 (55%)	4 (5.33%)	9 (45%)	3 (4%)	7 (35%)	1 (1.33%)
1	3 (15%)	7 (9.33%)	2 (10%)	6 (8%)	2 (10%)	2 (2.67%)
2	1 (5%)	12 (16%)	2 (10%)	10 (13.33%)	3 (15%)	8 (10.67%)
3	5 (25%)	52 (69.33%)	7 (35%)	56 (74.67%)	8 (40%)	64 (85.33%)
**Vascular** **morphology**						
None	11 (55%)	4 (5.33%)	9 (45%)	3 (4%)	7 (35%)	1 (1.33%)
Dotted or liner	7 (35%)	40 (53.33%)	8 (40%)	28 (37.33%)	6 (30%)	15 (20%)
Branch	1 (5%)	9 (12%)	1 (5%)	15 (20%)	3 (15%)	20 (26.67%)
Irregular	1 (5%)	22 (29.33%)	2 (10%)	29 (38.67%)	4 (20%)	39 (52%)
**Peripheral annular blood flow**						
None	20 (100%)	56 (74.67%)	20 (100%)	50 (66.67%)	19 (95%)	32 (42.67%)
Annular	0	19 (25.33%)	0	25 (33.33%)	1 (5%)	43 (57.33%)

Comments: Benign tumors: CDFI&PDI: χa2 = 1.067, pa = 0.785; χb2 = 0.600, pb = 0.896; χc2 = 0.000, pc = 1.000; MFI&CDFI: χa2 = 2.781, pa = 0.427; χb2 = 0.517, pb = 0.915; χc2 = 1.026, pc = 0.311; MFI&PDI: χa2 = 3.766, pa = 0.288; χb2 = 2.202, pb = 0.531; χc2 = 1.026, pc = 0.311. Malignant tumors: CDFI&PDI: χa2 = 0.550, pa = 0.908; χb2 = 4.721, pb = 0.193; χc2 = 1.158, pc = 0.282; MFI&CDFI: χa2 = 6.619, pa = 0.085; χb2 = 22.074, pb = 0.000; χc2 = 15.836, pc = 0.000; MFI&PDI: χa2 = 3.756, pa = 0.289; χb2 = 7.115, pb = 0.068; χc2 = 8.716, pc = 0.000. Where a represents Adler classification, b represents vascular morphology, c represents Peripheral blood flow.

**Table 5 jcm-12-00566-t005:** Diagnostic result of CDFI, PDI and MFI for renal tumors of different sizes (%).

Largest Diameter	Method	Sensitivity	95%CI	Specificity	95%CI	+LR	−LR	Positive predictive Value	Negative Predictive Value	Accuracy	AUC	Cutoff
**≤4 cm**	**CDFI**	72.09%	61.4~81.2	66.67%	41~86.7	2.16	0.42	91.17%	33.33%	71.15%	0.694	0.3876
	**PDI**	73.26%	62.6~82.2	66.67%	41~86.7	2.2	0.4	91.3%	34.29%	72.12%	0.700	0.3992
	**MFI**	93.02%	85.4~97.4	61.11%	35.7~82.7	2.39	0.11	91.95%	64.71%	87.5%	0.771	0.5413
**>4 cm**	**CDFI**	85.33%	75.3~92.4	75%	50.9~91.3	3.41	0.2	92.75%	57.69%	83.16%	0.802	0.6033
	**PDI**	89.33%	80.1~95.3	70%	45.7~88.1	2.98	0.15	91.78%	63.63%	85.26%	0.797	0.5933
	**MFI**	96%	88.8~99.2	65%	40.8~84.6	2.74	0.06	91.14%	81.25%	89.47%	0.805	0.61

## Data Availability

Available if requested.

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
