# Peer review of "Diagnostic Value of Multi-Mode Ultrasonic Flow Imaging Examination in Solid Renal Tumors of Different Sizes"

_jcm, 2023, doi:10.3390/jcm12020566_

Round 1
Reviewer 1 Report
Please see the attachment.

Author Response
请参阅附件。

Reviewer 2 Report
Dear authors,
to develop the diagnostic comparison, ROC curve is not enough. You should compute Sensitivity, Specificity, Positive likelihood Ratio, Negative likelihood Ratio, Positive predictive Value, Negative Predictive Value, Accuracy, and Kappa statistics where applicable.
Moreover, you should put more information in the introduction about "what is known", "what is not known", and "how your research can improve the scientific literature".
Round 2
Reviewer 2 Report
My suggestions have been solved.